# Experiences of Using Misoprostol in the Management of Incomplete Abortions: A Voice of Healthcare Workers in Central Malawi

**DOI:** 10.3390/ijerph191912045

**Published:** 2022-09-23

**Authors:** Bertha Magreta Chakhame, Elisabeth Darj, Mphatso Mwapasa, Ursula Kalimembe Kafulafula, Alfred Maluwa, Grace Chiudzu, Address Malata, Jon Øyvind Odland, Maria Lisa Odland

**Affiliations:** 1Department of Public Health and Nursing, Norwegian University of Science and Technology, 7491 Trondheim, Norway; 2School of Maternal, Neonatal and Reproductive Health, Kamuzu University of Health Sciences, Blantyre 312225, Malawi; 3Department of Research and Postgraduate Outreach, Malawi University of Science and Technology, Thyolo 310106, Malawi; 4School of Health Systems and Public Health, Faculty of Health Sciences, University of Pretoria, Pretoria 0002, South Africa; 5Department of Obstetrics and Gynaecology, St. Olav’s Hospital, 7030 Trondheim, Norway; 6Malawi-Liverpool-Welcome Trust Research Institute, Blantyre 312225, Malawi; 7Institute of Life Course and Medical Sciences, University of Liverpool, Liverpool L7 8TX, UK

**Keywords:** misoprostol, incomplete abortion, service providers, experiences, post-abortion care

## Abstract

Complications after abortion are a major cause of maternal death. Incomplete abortions are common and require treatment with surgical or medical uterine evacuation. Even though misoprostol is a cheaper and safer option, it is rarely used in Malawi. To improve services, an intervention was performed to increase the use of misoprostol in post-abortion care. This study explored healthcare providers’ perceptions and experiences with misoprostol in the Malawian setting and their role in achieving effective implementation of the drug. A descriptive phenomenological study was conducted in three hospitals in central Malawi. Focus group discussions were conducted with healthcare workers in centres where the training intervention was offered. Participants were purposefully sampled, and thematic analysis was done. Most of the healthcare workers were positive about the use of misoprostol, knew how to use it and were confident in doing so. The staff preferred misoprostol to surgical treatment because it was perceived safe, effective, easy to use, cost-effective, had few complications, decreased hospital congestion, reduced workload, and saved time. Additionally, misoprostol was administered by nurses/midwives, and not just physicians, thus enhancing task-shifting. The results showed acceptability of misoprostol in post-abortion care among healthcare workers in central Malawi, and further implementation of the drug is recommended.

## 1. Introduction

Maternal mortality in Malawi is among the highest in the world, with 439 deaths per 100,000 live births [1], and both spontaneous and induced abortions contribute to 6–7% of the maternal deaths [2]. Approximately 185 deaths per 100,000 abortions occur in Sub-Saharan Africa, and the mortality is high in countries where unsafe abortion is expected due to restrictive abortion laws [3]. In Malawi, abortion is only legal to save a woman’s life, but it has been estimated that around 140,000 abortions are performed every year [4,5]. Hence, most abortions in the country are likely unsafe [6]. Malawi has one of the highest rates of abortion complications in the world, and approximately 26,634 women seek post-abortion care in Malawian hospitals annually [6]. Treating abortion complications is a significant financial burden for the national health system [7], and unsafe abortions contribute significantly to maternal mortality in the country. The most common complication after abortion (induced and spontaneous) is retained products of conception, which is referred to as an incomplete abortion. Incomplete abortions can lead to bleeding, shock, sepsis, and potentially death if left untreated [8]. The situation needs to be treated with uterine evacuation, which can be achieved surgically or medically [9]. In Malawi, surgical treatment with sharp curettage is mainly used [10]. This is despite the World Health Organization’s (WHO) recommendation for the use of vacuum aspiration and medical treatment with the synthetic prostaglandin E_1_ analogue misoprostol in the management of incomplete abortions up to 14 weeks [9,11]. Misoprostol binds to smooth muscle cells in the uterine lining (myometrial cells) causing strong contractions. The prostaglandin also degrades collagen in the connective tissue and decreases cervical tone due to increased amplitude and frequency of contractions. These lead to the expulsion of retained products of conception from the uterus. [12,13] Common side effects are abdominal pain and diarrhoea which result from exposure to misoprostolic acid released during metabolism. Shivering and fever are side effects related to the effect of prostaglandin on the hypothalamus. In addition, nausea and vomiting, headache, and constipation may also occur. These side effects of misoprostol are self-limiting [14]. In many parts of the world, there has been a transition from surgical to medical treatment [15]. Still, Malawi as a country has not fully embraced the use of medical management of incomplete abortion. In a previous study from Malawi, only 1.4% of patients in three study hospitals in the southern region of the country were managed using misoprostol, and the rest of the cases were treated surgically [16]. The slow transition in the country can be attributed to limited availability of the drug, little staff knowledge on the use of misoprostol, fear of loss to follow-up, and reluctant physicians due to uncertainty of the outcomes [16]. Access to post-abortion care and family planning is essential to prevent unwanted pregnancies, unsafe abortions, and maternal deaths in countries with restrictive abortion laws. Post-abortion care is the management offered to a woman following abortion, and it consists of treatment of residual side effects, follow-up, and family planning [9]. Previous attempts to increase the use of manual vacuum aspiration (MVA) in Malawi illustrated issues, [10,16,17] and scarcity of MVA equipment, broken equipment, lack of trained personnel, and shortage of staff have made healthcare workers return to using sharp curettage in the management of incomplete abortions [6,10,18]. To ensure accessibility of quality post-abortion care (PAC) services to all women in need, it is necessary to scale up the use of misoprostol in Malawi. An intervention study was done in Malawi where healthcare workers from three public hospitals were trained on the use of misoprostol in the management of incomplete abortion with an aim of increasing usage of the drug. Health care workers were provided with training on the usage of misoprostol in PAC. Literature has shown that misoprostol is a safe and effective alternative to surgical treatment for first-trimester incomplete abortions [19,20]. This treatment has been proven cost-effective, non-invasive, and simple to use in both low- and high-income countries [15,19,21]. Healthcare workers are paramount in providing evidence-based quality PAC services [22]. To successfully implement medical treatment of incomplete abortions, there is a need to understand healthcare workers’ (physicians (clinical officers and doctors) and nurse/midwives) experiences and perceptions of using misoprostol before further scaling up the implementation. This qualitative study was carried out to explore healthcare workers´ experiences and perceptions of misoprostol use in the management of first-trimester incomplete abortion in three public hospitals in central Malawi.

## 2. Materials and Methods

### 2.1. Study Design

A descriptive phenomenological study using an inductive approach was performed. This qualitative study was nested within an intervention study aimed at increasing the use of misoprostol in the treatment of incomplete abortions. The intervention was conducted in three hospitals in July 2020; two hospitals served as controls. In the three intervention sites, women who presented with first-trimester incomplete abortion were treated with misoprostol 400 mcg sublingual or 600 mcg orally. The women gave information about their last menstrual period, were examined, and those in the first-trimester were given the drug. They were treated as outpatients if they were from near the hospital. Those from far waited for four to six hours at the clinic and then returned home. A follow-up visit was planned after one week. To explore the healthcare workers’ experience and attitudes towards using misoprostol, focus group discussions (FGD) were conducted, one at each of the three intervention sites.

### 2.2. Study Setting

The study was performed in three district hospitals of Mchinji, Salima and Bwaila in the central region of Malawi. The three hospitals were randomly selected as intervention sites for the main study on the use of misoprostol in the management of incomplete abortions.

### 2.3. Study Participants

The FDGs were conducted in August 2021, one year after the intervention. Purposive sampling was used to identify healthcare workers working in gynaecology. A total of 29 participants participated in the three FDGs. There were ten participants from Salima, nine from Mchinji and ten from Bwaila hospitals. Each of the participants participated in one focus group discussion.

### 2.4. Data Collection and Management

A topic guide developed by the research team was used to collect data from the healthcare workers on their experiences in the use of medical treatment for incomplete abortions. The study objective was the basis of the guide, and it had exploratory questions to guide the discussions. The guide was pretested before the actual data collection to determine the appropriateness regarding clarity and flow of questions. Three focus group discussions were conducted in English, one at each facility. The groups had a range of nine to ten participants. On average, the focus group discussions took one hour and 15 min. The discussions were audio recorded and transcribed verbatim immediately after the interviews.

### 2.5. Patient and Public Involvement Statement

The directors of health and social services and research committees in the hospitals were involved at the proposal stage before submission of the protocol for ethical review. The proposal was presented, and the research committee and the health and social services directors gave in their input. A focal person, either a nurse midwife or physician, was used as a gatekeeper to identify suitable participants. The results of this study will be shared with the ministry of health and the Directors of health and social services in all study sites.

### 2.6. Data Analysis

Thematic analysis was employed to analyse the data. The transcribed data were read several times before coding. Key ideas in the transcripts were noted, and coding was done manually based on the collected data. A codebook was developed where each code was defined, and decision rules on how to use the codes were laid down. After coding, codes were organised into subthemes which were later organised into themes. The last author checked the codes and themes. Part of the analytical process is shown in Figure 1. The data are presented in narratives.

Data collection was done by the first author collectively with two research assistants who had never worked in the study sites and had no prior relations with the participants. This helped to maintain neutrality.

### 2.7. Ethical Considerations

Ethical approval was obtained from the college of medicine research and ethics committee (COMREC)-Malawi (P.01/20/2924) and REK-Norway (141130 2019). Permission to conduct the study in the selected sites was obtained from Lilongwe (for Bwaila), Salima, and Mchinji district health offices. During data collection and management, codes were used for anonymity. Before data collection, participants were given information sheets about the study. After becoming familiar with the information, they provided written consent. The participants were informed of the study’s potential risks and benefits. They were assured of privacy and confidentiality, as no names were used for their identification. Quotes were coded indicating respondents, e.g., [res 1B] or [res 6S]. Participants were informed that participation in the study was voluntary and that they were free to withdraw from the study at any point. The data were stored safely and accessed only by the research team.

## 3. Results

### 3.1. Demographic Characteristics of Participants

Three FDGs were conducted in the three public hospitals where the intervention was implemented. A total of 29 healthcare workers aged 24 to 49 were enrolled as participants. Ten participants were from Salima (S), nine from Mchinji (M) and ten from Bwaila (B) hospitals. The group comprised of clinical officers and nurse midwives working in gynaecological wards. Following thematic analysis, two themes emerged, which had six subthemes each (Table 1). These are presented in detail, along with some of the participants’ own words.

### 3.2. Facilitators and Barriers to the Use of Misoprostol for First-Trimester Incomplete Abortion

#### 3.2.1. Health Providers’ Knowledge of the Use of Misoprostol for First-Trimester Incomplete Abortion

Appropriate knowledge among healthcare workers is key to implementing health services. Many participants indicated that they had knowledge of the use of misoprostol in the management of incomplete abortion. The level of knowledge ranged from very knowledgeable to partly knowledgeable. When asked about the source of knowledge on the use of misoprostol in the management of incomplete abortion, some participants indicated that they had undergone formal training. In contrast, others learnt on the job and others got the information from the literature. The healthcare workers further indicated that though they may be knowledgeable about managing women using misoprostol, there is a need for more formal training to equip staff with knowledge and skills in medical management.


*“It was from school I did not go for the [in-service] training, and also … on job training, we get if from friends, those people who know the management, sometimes they can also post the management in the wards so we just read from there”.*
[res 1B]


*“We may have knowledge on how to use misoprostol in the first-trimester, some just read articles but haven’t gone through a formal training. So, we need a lot of people to be trained. And also, as an institution not all of us will [always] be here, some will be coming [and] some will be going out so we have a lot of newcomers who don’t have knowledge on how to use it. As long as we are doing mentorship, but also trainings are much important”.*
[res 9 S]

#### 3.2.2. Health Providers’ Confidence in the Use of Misoprostol

The healthcare workers felt that they were confident in using misoprostol in the management of first-trimester incomplete abortion. The high confidence level was attributed to the knowledge they acquired during training, support from fellow staff and the reference materials made available to them.


*“I am also confident when am treating women who have undergone incomplete abortion because I have all the knowledge that is required to treat these women using medical management, and whenever I am stuck I also go back to my notes because I have a handout that you gave us last time, just to make sure that I am doing the right thing”.*
[res 1 M]

Other healthcare workers indicated that they were not confident with the treatment because they feared treatment failure, while others lacked confidence because they lacked knowledge.

#### 3.2.3. Experienced Benefits of Misoprostol in the Management of First-Trimester Incomplete Abortion

Overall, healthcare workers welcomed the use of misoprostol in post-abortion care. Several benefits in using misoprostol were noted. When asked about their experiences on the effectiveness of misoprostol following first-trimester incomplete abortion, almost all healthcare workers indicated that they perceived the drug as effective. They estimated that about 80 to 90% of the patients had a complete abortion after treatment. However, misoprostol was seen to be less effective by a few healthcare workers, especially in women who were unsure of their gestational age and those who gave wrong dates. Some healthcare workers indicated that misoprostol is only effective in evacuating the uterus when administered correctly. In other words, it is effective if given to the right patient, one with first-trimester incomplete abortion and with the correct dosage and route.


*“I can say that the misoprostol was effective, because for the women who came… with the right information about the gestational age they were doing fine at the review day, no bleeding no whatever, during scanning everything is complete. But, for the less that were giving wrong information of gestational age …, they were coming again with bleeding and after scanning you find that there is still remains of products of conception and after further investigations you find out that they were giving wrong information on gestation age, maybe it was over three months but for the first-trimester [incomplete abortions] it was really effective”.*
[res 8 S]

It was also revealed that misoprostol was cost-effective in post-abortion care as less staff and equipment were used with minimal client contact period. In addition, it was seen to be safe, simple and easy to use. Misoprostol was reported to reduce the chances of complications such as sepsis and injuries. It was further said that the use of misoprostol reduced the level of anxiety experienced by patients.

Another commonly acknowledged benefit was that the use of misoprostol in first-trimester incomplete abortions further reduced the workload on healthcare workers, as congestion in the gynaecological wards was reduced. Less time was required to be with the patient, as women could be discharged immediately after treatment or after a few hours of observation hence reducing the patient’s hospital stay. Only one participant pointed out that using misoprostol wasted time, especially in cases where it failed and the patient still required surgical management.


*“The advantages were that, … misoprostol takes less time, reduces workload for us health workers and also congestion in the ward were reduced. And also, for the patients themselves they stay less time in the hospital. Maybe for the MVA some of them would wait for two days before the MVA was done. But for this the same day they came after history taking, examination, and find out they are eligible for the medical management, they were given and go home and also less complications than the MVA. It doesn’t need invasive procedures that means less injuries due to the MVA or curettage and also reduces infection”.*
[res 8 S]

Another reported benefit of misoprostol in post-abortion care was that it provided privacy to patients. Moreover, administering misoprostol did not require advanced training for healthcare workers.


*“Just to add it also provides privacy, … some people with abortion, for others to know they have gone through abortion is like they don’t feel comfortable, so with this misoprostol they can just be treated at OPD [outpatient department] and go home…And in addition, when giving misoprostol you just need to know how you are going to give the misoprostol unlike MVA whereby you have to go through training in order to provide the service, you can just even read the notes on how to give misoprostol you are good to go.…… Like in my case I was not trained, during that time I was on… COVID isolation, … but when I came back I read the notes and I am able to give it”.*
[res 8 M]

#### 3.2.4. Availability and Accessibility of Misoprostol for First-Trimester Incomplete Abortion

When addressing questions on availability and accessibility of the drug. It was noted that mostly misoprostol was available and accessible in the study intervention sites. The drug was available at the pharmacy and was easily accessible for the healthcare workers. In one hospital, the stocks at the pharmacy were supplemented by stock kept at the matron’s office. This made it easy to access misoprostol even when the drug was out of stock at the pharmacy. The demand for the drug was seen to have increased during the study period, and this led to drug unavailability at times.


*“Since the introduction of this medical management of incomplete abortion using misoprostol, the demand now is very high so we don’t always have this drug because we encounter so many clients per day and also our pharmacy does not always have this drug, so sometimes we also experience this challenge of being out of stock”.*
[res 1 M]

At one facility, it was reported that the drug is only prescribed by physicians to prevent drug misuse. Hence, this made it inaccessible to the nurse midwives. Others indicated drug unavailability and measures to control drug use as barriers to its use. In addition, fear of misuse was communicated.

#### 3.2.5. Health Providers’ Satisfaction with the Use of Misoprostol in Post-Abortion Care

Most healthcare workers expressed satisfaction with misoprostol as they found it efficient with satisfactory outcomes.


*“Yes, we are very much satisfied with the use of misoprostol in managing incomplete abortion as we have already said it doesn’t require much time…..”.*
[res 3 S]

Despite these expressions, mixed reactions on levels of satisfaction were noted. Some participants said they were not fully satisfied, while others indicated that they were very satisfied with the use of the drug. Not honouring follow-up visits by the patients was a challenge that made healthcare workers uncertain about the outcome of the management and hence did not feel satisfied with medical management. Others reported that the drug was not always effective as some patients came back with retained products of conception, and others developed complications.

#### 3.2.6. Supportive Supervision in the Use of Misoprostol

Participants lamented the lack of supportive supervision in using misoprostol as a big challenge in the provision of care.


*“We can say that supervision is not there … No, we assume that the person who is taking the drug knows how to use it, know how to handle it”.*
[res 5 S]

### 3.3. Care Provided to Women with First-Trimester Incomplete Abortion

#### 3.3.1. An Encounter with a Patient: Treatment Offered

The healthcare workers shared their experiences in caring for women with first-trimester incomplete abortion. It was indicated that women are assessed first, and when gestation is established to be within the first-trimester or that the retained products of conception are minimal, then medical management is considered. If the gestational age is beyond the first-trimester and/or if the woman has more retained products of conception shown after an ultrasound scan, they opt for MVA. The women are also offered family planning services as part of management.


*“…the patient may present with bleeding following… few months amenorrhea. … the patient is… tested for pregnancy…, submit urine… we… check if patient was really pregnant. …when the test comes… positive, we… proceed to… do… abdominal examination and assess gestational [age in] weeks. …if … gestation is… within… first-trimester, the woman [is] treated with…misoprostol… 600 mcg orally, and then we observe her… for… few hours in case she might have… complications like… bleeding if she has no… complication we… explain to her… that she might experience… back pain and some bleeding as the drug… clean… the uterus, we… explain… to the woman so that she should not wonder when it happens. …we advise her to report any danger signs like … signs of shock, if she bleeds excessively she has to report back. …we give her …one week …. for review, [to] see if the drug has been successful when she comes… we take history… about … her experience post drug. So, if… no… complications, we proceed… giving her the post-abortion family planning … of her choice…”.*
[res 2 M]

#### 3.3.2. Preferred Treatment by Healthcare Workers

Almost all participants cited that they would prefer misoprostol over MVA in managing first-trimester incomplete abortion except one participant who preferred MVA. Some benefits of misoprostol were cited as reasons for choosing medical treatment over surgical treatment. Among the benefits mentioned as reasons for preferring medical treatment were: cost-effectiveness, simplicity to use, less time with the patient, and being less invasive. Some healthcare workers preferred misoprostol because its use addressed some of the challenges they faced with MVA, such as lack of training and equipment.


*“I will prefer using misoprostol since it will not require much time like it will take doing the MVA because you need to prepare the patient and all the equipment you assemble … with misoprostol you just take the medication and give the client and she can go home right away”.*
[res 3 S]

On the other hand, one participant was unconvinced that misoprostol is efficient and preferred MVA over the drug. The lack of guarantee of a complete evacuation of the uterus contributed to the preference of MVA since a completed MVA assures that the uterus is completely evacuated.

#### 3.3.3. Experiences of Follow-Up Care of Patients Treated with Misoprostol

Follow-up care is vital to ensure that patients are doing well after treatment. It was observed that some patients come back after a week for follow-up but others do not. Some patients feel that they are healthy and treated and see no point or need to report back to the hospital, but almost all those who experience problems return.


*“To be honest I think those who come are those who have problems because when we treat these women if they are well they don’t come back, but if they feel like something else is not working on them, they come back maybe even [before] the day you have given them. So, we can say that during follow-up not more are coming…”.*
[res 2 B]

#### 3.3.4. Complications

When asked about the complications women developed after receiving misoprostol, most participants had a common response: that the women did not develop complications. Only two participants had patients who reported back to the clinic with complaints of fever and abnormal vaginal discharge. Administration of the drug to women beyond the first-trimester was regarded as the genesis of the complications.


*“Like for me, I have never seen a woman who has experienced a major complication following the treatment …using misoprostol. They only complain about pain, … others … complain about bleeding of which we already expect … so as for me I have never seen any major complication”.*
[res 1 M]

#### 3.3.5. Type of Healthcare Workers Providing Medical Management

In response to the question on who provides the service to women. It was discovered that nurse midwives, obstetricians, clinical officers and doctors prescribed and provided misoprostol to women at two facilities. At the third facility, nurse midwives were not allowed to prescribe the drug, but they were allowed to administer the drug to patients. This was for control purposes as there was fear of drug misuse.


*“The challenge is that we [are] … not really practicing it. Most [of the] time we depend upon clinicians [physicians] to order and give the drug. But we know how to give it and manage the patient… We give but under clinician’s prescription because when documenting always the pharmacy people wants the name of the clinician who prescribes”.*
[res 6 S]

#### 3.3.6. Task Shifting

The study results also revealed that with misoprostol, there was task shifting as the work which was primarily dependent on physicians could also be done by nurse midwives.


*“Yah there is task shifting because nurses as well… can handle these patients. During some weekends maybe, some clinicians might not be available, the nurses can do this management”.*
[res 2 M]

## 4. Discussion

The study results revealed that misoprostol is welcomed by healthcare workers. Many healthcare workers reported that they were knowledgeable and confident in using misoprostol. Formal training, literature and learning from colleagues on the job were the sources of information about the drug. This study also found that the healthcare workers preferred misoprostol because it is considered safe, effective, easy to use, cost-effective, has fewer complications, reduces congestion in hospitals, reduces workload, and saves time. Obstetricians, clinical officers and nurse midwives provided misoprostol to women with incomplete abortions and this enhanced task shifting. This is particularly important in settings where staff shortage is a problem. The healthcare workers expressed satisfaction with the outcomes of treatment. These results showed acceptability of misoprostol in the management of first-trimester incomplete abortion by the healthcare workers on the basis of their experiences [20].

Knowledge is vital in the implementation of healthcare services. Most participants in this study reported that they were knowledgeable and confident to use misoprostol. However, some healthcare workers reported being less confident with the treatment because they were not trained to use misoprostol and feared treatment failure. It was further indicated that though health providers may have knowledge about the management of women using misoprostol, there is still a need for more formal training to equip them with knowledge and skills in medical management. Healthcare worker training is vital in empowering health personnel with new knowledge and skills and will improve access to post-abortion care services [23]. There is a need to implement initiatives such as training middle-level healthcare workers such as nurse midwives and clinical officers to support the implementation of PAC in LMICs [24]. These pieces of training will promote task shifting as work that would previously wait for a physician can be done by nurse midwives who are always available. Involving a wider range of providers is a strategy that would help reduce maternal morbidity and mortality in LMICs [24,25].

Task shifting is paramount in facilities where there is a shortage of staff, and it helps to improve the accessibility of services. The results found that clinical officers, doctors, nurse midwives and obstetricians were all administering misoprostol, except for one facility where nurse midwives were not allowed to prescribe the drug. System barriers to uptake of PAC, such as preventing mid-level providers from the provision of PAC and fears of providing the treatment due to restrictive policies, were also reported in a systematic review done in the sub-Saharan Africa [24]. In Gabon, it was reported that restricting midwives from managing women with incomplete abortion increased the waiting time, and it raised the rate of complications from 2% to 14.1% [26]. These findings necessitate empowering nurse midwives to provide PAC services to prevent complications and promote quality of life among women. Task shifting in the two public hospitals was noted as more nurse midwives were able to manage the patients independently. Studies have shown that trained nurse midwives can provide misoprostol for PAC with the same effectiveness as physicians [21,24,25]. Task shifting can help reduce the shortage of staff and workload in hospitals, thereby reducing delays in the initiation of treatment.

Misoprostol was reported to be safe and effective, most women did not develop complications, and only two women treated with misoprostol were reported to have developed sepsis within the year of intervention. This added to the evidence on the safety of the drug when used correctly, as reported by researchers previously [19]. In studies done in Nigeria, Senegal and Spain, it was also found that misoprostol was as effective as MVA in the management of first-trimester incomplete abortion [19,20,25]. The development of infection in this study was related to a gestation age of more than 12 weeks. To address the problem of administering the drug in women whose gestational age is above the recommended, there is a need to provide training to healthcare workers on proper gestational age assessment.

In addition, the healthcare workers preferred misoprostol over surgical management as it was perceived to be cost-effective, simple to use, less invasive and satisfactory to staff [23]. The cost-effectiveness of misoprostol was also reported by Cubo et al. and Benson et al. [15,19]. In low-income countries like Malawi, access to post-abortion care is a priority. Due to challenges faced in using MVA, such as broken equipment and lack of trained staff, there is limited access to PAC services [24]. In the current study, the use of misoprostol addressed these challenges as the provision of PAC required no equipment and no advanced training. It was further reported that the use of the drug reduced the workload of staff as its use required less time with the patient and reduced congestion in the wards. A study by Cubo et al. found evidence that misoprostol is cost-effective in LMICs [19], as the women can take it at home or as outpatients. This can help relieve some of the burden on the healthcare system that is faced with challenges because of the scarcity of equipment and human resources capable of attending to these women and congestion in hospitals [18]. The use of misoprostol will reduce problems of staff shortages, consequently reducing waiting time at the hospital for women to receive treatment. Reduced waiting time will decongest hospitals, and use of misoprostol will ease the financial burden as it is cost-effective and will save money for hospitals. An average save of EUR 1576.80 per client was reported by Cubo et al. after comparing misoprostol and curettage [19]. In a study done in Malawi, it was found that the use of MVA and misoprostol would produce an estimated cost reduction of PAC by 30 % [15].

The drug was reported to be available in the intervention sites most of the time, but there were times when misoprostol was out of stock. The situation of stockouts is similar to what was found in Zimbabwe, where 55% of health facilities reported stockouts of misoprostol [27]. Drug stockouts hinder the effective provision of post-abortion care in the facilities and lead to poor quality of services [24]. There is a need for hospital management to ensure a constant supply of the drug for effective service delivery. Another factor that hinders service provision is the lack of supportive supervision. All healthcare workers reported that no supportive supervision was provided to them. Constant supervision helps improve self-confidence, which in turn helps one gain proficiency in service provision. There is a need for hospital management to enhance supportive supervision leadership to staff. The possible barriers reported in our study are different from those found in Nigeria, where preference to use MVA and providers not being sure of the brand of misoprostol were the main barriers [28].

Follow-up care is paramount in ensuring that the treatment is effective. In this study, it was further revealed that some women came back for follow-up, but others did not once they felt that they were fine. This was worrisome to the healthcare workers as they thought the women might develop complications at home. The healthcare workers would like all women to report back after a week for follow-up, especially during treatment introduction, to ensure patient safety. This calls for community sensitisation on the importance of follow-up.

### Strength and Limitations

Strengths of this study were that it was conducted in three different districts in the central region of Malawi, and various types of healthcare workers (nurse midwives and physicians) were involved. Data collection was done by people with no prior relations with the participants, which helped maintain neutrality. To ensure reflexivity, care was taken that the authors’ experience did not have an impact on the data collection and analysis process, and checking of code and theme development was done between authors to ensure that previous experience did not influence data presentation. The study used a qualitative method, so the findings are not aimed for generalisation to other settings. Describing the study’s context, participants, study design, research method and data analysis aim to enhance the study’s trustworthiness.

## 5. Conclusions

Evidence indicates that misoprostol is an effective treatment for incomplete abortions. The emergent findings show that misoprostol is acceptable and preferred by many healthcare workers in the central part of Malawi because it is perceived as safe, easy to use, effective and time-saving. The use of misoprostol in Malawi could increase the accessibility and quality of PAC services currently offered. To enhance its use in PAC, further training interventions and supportive supervision of healthcare workers are recommended.

## Figures and Tables

**Figure 1 ijerph-19-12045-f001:**
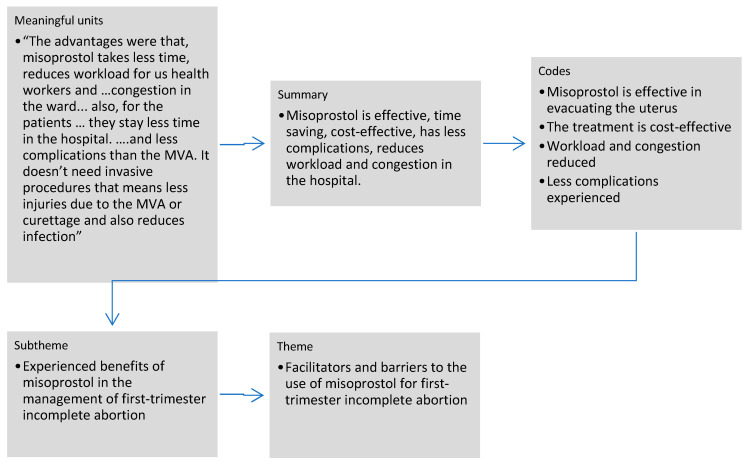
Analytical process.

**Table 1 ijerph-19-12045-t001:** Themes and subthemes generated from FDGs of healthcare workers on their experiences in the use of misoprostol in first-trimester incomplete abortion (Malawi-2).

Theme	Subthemes
Facilitators and barriers to the use of misoprostol for first-trimester incomplete abortion	Health providers’ knowledge of the use of misoprostol for first-trimester incomplete abortion
	Health providers’ confidence in the use of misoprostol
	Experienced benefits of misoprostol in the management of first-trimester incomplete abortion
	Availability and accessibility of misoprostol for first-trimester incomplete abortion
	Health providers’ satisfaction with the use of misoprostol in post-abortion care
	Supportive supervision in the use of misoprostol
Care provided to women with first-trimester incomplete abortion	An encounter with the patient: treatment offered
	Preferred treatment by the healthcare workers
	Experiences of follow-up care of patients treated with misoprostol
	Complications
	Type of healthcare workers providing medical management
	Task shifting

## Data Availability

Data would be made available upon reasonable request from the corresponding author.

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
