# Peer review of "Experiences of Using Misoprostol in the Management of Incomplete Abortions: A Voice of Healthcare Workers in Central Malawi"

_ijerph, 2022, doi:10.3390/ijerph191912045_

Round 1

Reviewer 1 Report

The study is simple and clear, the findings are convincing. Nonsurgical intervention is better than surgical for the patient and for the health care staff. To the extent that this study might facilitate further implementation of the drug, the study has value.

One slight problem is the introduction lists 'little staff knowledge' as one of the reasons this treatment has not been more fully adopted, but the findings do not show this. It is implied that the researchers themselves provided training at an earlier time. If this is the case, it should be stated. 

I wondered if the drug company had any role in funding the project. 

Author Response

A point-by-point response to the reviewer

Reviewer 1

Point 1:

One slight problem is the introduction lists 'little staff knowledge' as one of the reasons this treatment has not been more fully adopted, but the findings do not show this. It is implied that the researchers themselves provided training at an earlier time. If this is the case, it should be stated. 

Response: Thank you for the comment, an intervention study was done where healthcare workers were trained on the use of misoprostol in the treatment of incomplete abortion with an aim of increasing the use of the drug. The current study was part of an evaluation of the project. It was an oversight that it was not mentioned. A revision has been made and the information has been added.

“An intervention study was done in Malawi where healthcare workers from three public hospitals were trained on the use of misoprostol in the management of incomplete abortion with an aim of increasing usage of the drug. Health care workers were provided with training on usage of misoprostol in PAC.”

Please see page 2 lines 79 to 82

Point 2:

I wondered if the drug company had any role in funding the project. 

Response: Thank you for the comment. A drug company had no role in funding the project.

Reviewer 2 Report

The Ms entitled Experiences of using misoprostol in the management of incomplete

abortions: A voice of healthcare workers" is of relevance to the audience since highlight a social/medical problem. The Ms is easy to follow and pleasant to read. However to help the reader who is not so competent in the field. For example a brief summary of the mechanisms of action of misoprostol should be added. I would like to suggest to briefly discuss potential side effects, either already known by clinical evidence or potentially present due to differential effects of pge1. Treatment with  PGE1 provokes an elevation of uterine tone and a long-lasting sustained contraction. It could be of interest to report previous finding indicating that the effects of PGE1 on the uterus cannot be solely explained by  increased myometrial contractions but also by a decreased collagen content (Am J Perinatol 2012 Sep;29(8):615-22). This effect could be of importance since female genital tissue morpho-histological alterations including those related to the collagen content are an important unedrlying mechanism of the genitourinary syndrome (often related to menopause but also to the premenopausel period) (sexMed Rev. 2018 Oct;6(4):558-571. doi: 10.1016/j.sxmr.2018.03.005. Epub 2018 Apr 7.)

Author Response

A point-by-point response to the reviewer

Reviewer 2

Point 1:   

To help the reader who is not so competent in the field. For example, a brief summary of the mechanisms of action of misoprostol should be added.

Response: Thank you for the comment, information on the mechanism of action of misoprostol has been added to the introduction.

“Misoprostol binds to smooth muscle cells in the uterine lining (myometrial cells) causing strong contractions. The prostaglandin also degrades collagen in the connective tissue and decreases cervical tone due to increased amplitude and frequency of contractions, this leads to expulsion of retained products of conception from the uterus.”

Please see page 2 lines 54 to 58

Point 2:

I would like to suggest to briefly discuss potential side effects, either already known by clinical evidence or potentially present due to differential effects of pge1. Treatment with  PGE1 provokes an elevation of uterine tone and a long-lasting sustained contraction. It could be of interest to report previous finding indicating that the effects of PGE1 on the uterus cannot be solely explained by  increased myometrial contractions but also by a decreased collagen content (Am J Perinatol 2012 Sep;29(8):615-22). This effect could be of importance since female genital tissue morpho-histological alterations including those related to the collagen content are an important unedrlying mechanism of the genitourinary syndrome (often related to menopause but also to the premenopausel period) (sexMed Rev. 2018 Oct;6(4):558-571. doi: 10.1016/j.sxmr.2018.03.005. Epub 2018 Apr 7.)

Response: Thank you for the comment, the information has been added to the introduction.

“Common side effects are abdominal pain and diarrhoea which result from exposure to misoprostolic acid released during metabolism. Shivering and fever are side effects related to the effect of prostaglandin on the hypothalamus. In addition, nausea and vomiting, headache, and constipation may also occur. These side effects of misoprostol are self-limiting.”

Please see page 2 lines 58 to 62
